# Drought Stress Alleviator Melatonin Reconfigures Water-Stressed Barley (*Hordeum vulgare* L.) Plants’ Photosynthetic Efficiency, Antioxidant Capacity, and Endogenous Phytohormone Profile

**DOI:** 10.3390/ijms242216228

**Published:** 2023-11-12

**Authors:** Neveen B. Talaat

**Affiliations:** Department of Plant Physiology, Faculty of Agriculture, Cairo University, Giza 12613, Egypt; neveen.talaat@agr.cu.edu.eg

**Keywords:** melatonin, water deficit stress, barley (*Hordeum vulgare* L.), antioxidant response, photosynthetic activity, phytohormone production

## Abstract

The production of crops is severely limited by water scarcity. We still do not fully understand the underlying mechanism of exogenous melatonin (MT)-mediated water stress tolerance in barley. This study is the first of its kind to show how MT can potentially mitigate changes in barley’s physio-biochemical parameters caused by water deficiency. Barley was grown under three irrigation levels (100%, 70%, and 30% of field capacity) and was foliar sprayed with 70 μM MT. The results showed that exogenously applied MT protected the photosynthetic apparatus by improving photosynthetic pigment content, photochemical reactions of photosynthesis, Calvin cycle enzyme activity, gas exchange capacity, chlorophyll fluorescence system, and membrane stability index. Furthermore, the increased levels of salicylic acid, gibberellins, cytokinins, melatonin, and indole-3-acetic acid, as well as a decrease in abscisic acid, indicated that foliar-applied MT greatly improved barley water stress tolerance. Additionally, by increasing the activity of antioxidant enzymes such as superoxide dismutase, catalase, ascorbate peroxidase, monodehydroascorbate reductase, dehydroascorbate reductase, and glutathione reductase and decreasing hydrogen peroxide content, lipid peroxidation, and electrolyte leakage, MT application lessened water stress-induced oxidative stress. According to the newly discovered data, MT application improves barley water stress tolerance by reprogramming endogenous plant hormone production and antioxidant activity, which enhances membrane stability and photosynthesis. This study unraveled MT’s crucial role in water deficiency mitigation, which can thus be applied to water stress management.

## 1. Introduction

Water scarcity poses a significant risk to global agriculture. Plants respond to water deficit stress by modifying photosynthetic pathways, regulating antioxidant responses, and changing plant hormone content [1,2,3]. Previous studies have shown that a lack of water causes an increase in reactive oxygen species (ROS) production, which causes oxidative damage to chloroplast and a decrease in chlorophyll content that limits photosynthesis [4,5]. Plant hormones such as indole-3-acetic acid, ethylene, jasmonic acid, gibberellic acid, cytokinins, abscisic acid, and salicylic acid are important signals for plant growth and development [6,7], and they also play a role in plant stress response [8,9,10,11]. To meet the growing demand of the world population by 2050, we will need to increase food crop production by 87% [12]. Exploring appropriate approaches to improve water stress tolerance in the barley plant is, therefore, critical to improving its growth and production.

A broad class of signaling molecules known as phytohormones is a potent regulator of plant growth and productivity [13]. Stressful circumstances can modify the level of endogenous hormones by regulating the biosynthesis or degradation of hormones either directly or indirectly [8,14]. Phytohormones are regarded as important endogenous molecules for regulating physiological and biochemical responses, even though the plant’s response to stresses is dependent on a variety of factors [10,11]. Numerous studies have demonstrated that MT provides significant benefits for plant growth and development under stressful conditions [5,15,16,17,18]. Therefore, there is growing interest in employing MT as an effective plant growth regulator to mitigate the harmful effects of water stress on plants.

Melatonin (MT, *N*-acetyl-5-methoxytryptamine) is an eco-friendly molecule with strong antioxidant properties that plays an important role in stimulating a variety of physiological and chemical responses to adverse environmental conditions [18,19,20,21]. Solid evidence suggests that MT is an effective anti-stress factor in plants [21,22,23,24]. Drought tolerance is induced via MT application by suppressing ROS accumulation and stabilizing biological membranes [3,15,25,26]. Furthermore, by activating antioxidant enzymes and raising photosynthetic electron transport efficiency, MT increases plant resistance to stress [11,27,28,29,30]. Notably, MT has been demonstrated in multiple experiments to improve redox homeostasis, cell membrane stability, and photosynthetic efficiency under a variety of abiotic stressors [24,26,29,31,32,33]. Comparably, exogenous MT reduces oxidative damage and shields photosystem II from the damaging effects of drought [34]. By increasing gas exchange capacity, chlorophyll content, Rubisco activity, and the expression of genes linked to photosynthetic processes, MT-mediated photosynthetic performance under heat stress [17]. According to recent research, exogenously applied MT also decreases the degradation of chlorophyll and increases the expression of genes related to proteins during the synthesis of chlorophyll [35].

Further evidence also showed that MT regulates stress responses in concert with other phytohormones [20,36,37]. Exogenous MT treatment has been shown to upregulate the expression of multiple genes associated with adventitious root formation and the auxin signaling pathway in tomato plants [38]. Actually, by controlling rooting via auxin induction and aerial production via an increase in cytokinin production, MT application can impact how plants respond to abiotic stress [9]. In response to various abiotic challenges, exogenous MT treatment controls the genes involved in the biosynthesis or catabolism of plant hormones, such as indole-3-acetic acid, cytokinins, gibberellins, abscisic acid, jasmonic acid, salicylic acid, and ethylene [9,39,40]. Through the upregulation of the cytokinin production genes during heat stress, MT application can raise the quantity of endogenous cytokinin [41]. In response to stress caused by drought [11,42], salt [43], and cold [44], MT treatment also controlled the levels of abscisic acid and gibberellins. Additionally, it has been noted that exogenous MT raised the endogenous content of indolyl-3-acetic acid and MT in salt-stressed plants [45]. Similarly, MT treatment increases plant resistance to dehydration-induced leaf senescence by controlling the networks of genes and phytohormones, mostly by increasing IAA concentration and decreasing ABA content [46]. Under water stress conditions, it was discovered that MT and other phytohormones like indole-3-acetic acid, gibberellin, cytokinin, jasmonic acid, salicylic acid, and abscisic acid interact and that many MT-affected genes take part in different hormone signaling pathways [6,11,26]. These findings suggested that MT’s ability to confer stress resistance may also be influenced by the control of hormone homeostasis.

Barley (*Hordeum vulgare* L.) is one of the most significant cereal crops farmed in Europe, the Middle East, North and South Africa, and Asia. Water scarcity is becoming an essential element influencing agricultural sustainability, and overcoming its injuries and impact on crop production is a serious challenge globally [1,5]. Considering MT’s anti-stress capabilities [19,23], the present piece of work was undertaken to explore the potential function of MT in enhancing water stress tolerance in barley based on changes in endogenous hormonal status, leaf photosynthetic efficiency, and cellular membrane stability. This could be the first study to look into whether MT treatment can boost barley’s ability to withstand water stress by influencing internal phytohormone production. Accordingly, it has been hypothesized that exogenous MT application may alleviate water stress injuries and prevent drought-induced growth and production inhibition in barley plants by modifying the endogenous phytohormone profile, photosynthetic activity, and antioxidant response. In this respect, the changes in the concentration of indole-3-acetic acid, gibberellins, cytokinins, abscisic acid, salicylic acid, and melatonin, along with the changes in the photosynthetic pigment concentration, photochemical reaction activity, gas exchange capacity, chlorophyll fluorescence system, Calvin cycle enzyme activity, hydrogen peroxide content, lipid peroxidation, antioxidant enzyme (superoxide dismutase, catalase, ascorbate peroxidase, monodehydroascorbate reductase, dehydroascorbate reductase, glutathione reductase) activity, membrane stability index, and electrolyte leakage were evaluated in barley leaves to highlight the exact mechanism of MT-induced water stress tolerance and consequently improved plant growth and productivity. Correlations between MT application and endogenous phytohormone production may offer a unique insight that can aid in promoting plant growth under stressful circumstances. Indeed, understanding how MT foliar spray regulates plant growth and development under water stress can greatly speed up their usage in crop protection and enhancement.

## 2. Results

### 2.1. Foliar-Applied MT Alleviates Water Stress-Induced Decreases in Barley Growth and Productivity

The obtained results revealed that both moderate (70% FC) and severe (30% FC) water stress conditions significantly inhibited barley growth and yield parameters in terms of plant height, leaves number, total leaf area, shoot dry weight, grains number, and grain yield. However, the foliar application of MT significantly alleviated the growth and productivity inhibition brought on by water deficiency (Figure 1A–F). In comparison to the untreated plants under well-watered and water-stressed (70% and 30% FC) conditions, respectively, it significantly (*p* < 0.05) enhanced plant height by 37.7%, 46.9%, and 65.8%, leaves number plant^−1^ by 38.6%, 45.0%, and 75.0%, total leaf area plant^−1^ by 48.8%, 61.2%, and 80.0%, shoot dry weight plant^−1^ by 53.4%, 73.3%, and 107.7%, number of grains plant^−1^ by 51.2%, 75.4%, and 110.9%, and grain yield plant^−1^ by 57.2%, 83.8%, and 130.8%.

### 2.2. MT Foliage Application Ameliorates the Inhibition of Photosynthetic Pigments Content in the Water-Stressed Barley Plants

Chlorophyll *a*, chlorophyll *b*, carotenoids, and total pigment concentrations were considerably lower in the stressed barley plants’ leaves than in the unstressed ones. In comparison to the control (100% FC), plants grown under 30% FC had the greatest reduction in the acquisition of chlorophyll *a*, chlorophyll *b*, carotenoids, and total pigments, with 34.5%, 55.0%, 59.5%, and 44.8%, respectively. However, MT foliage application significantly (*p* < 0.05) increased the leaf photosynthetic pigments concentration under non-stressed and stressed conditions (Figure 2A–D). In comparison to the untreated plants under well-watered and water-stressed (70% and 30% FC) conditions, respectively, it significantly increased the concentration of chlorophyll *a* by 32.5%, 45.9%, and 63.9%, chlorophyll *b* by 47.7%, 74.4%, and 154.4%, carotenoids by 59.5%, 116.7%, and 180.0%, and that of total pigments by 40.9%, 62.1%, and 100.5%.

### 2.3. Exogenously Applied MT Mitigates Water Deficiency-Induced Reduction in PSI and PSII Activities

The activities of PSI and PSII were significantly decreased under water deficiency. Contrarily, the MT foliage application dramatically increased their activity in comparison to the untreated plants (Figure 3A,B). Under well-watered and water-stressed (70% and 30% FC) conditions, it significantly (*p* < 0.05) increased the PSI activity by 37.4%, 58.9%, and 114.5% and the PSII activity by 49.5%, 108.3%, and 212.5%, respectively, in comparison to values of the untreated plants.

### 2.4. The MT Foliage Application Triggers Leaf Gas Exchange Attributes in the Water-Stressed Barley Plants

Severe water stress (30% FC) condition significantly (*p* < 0.05) reduced net photosynthetic rate (*P_n_*), stomatal conductance (*G_s_*), and transpiration rate (*T_r_*) of barley leaves by 53.8%, 67.3%, and 57.2%, respectively, compared to the normal irrigation. On the contrary, the exogenously applied MT significantly increased these variables relative to the stressed untreated plants (Figure 4A–C). Compared with the untreated plants, it significantly (*p* < 0.05) enhanced the *P_n_* by 32.9%, 55.1%, and 125.0%, *G_s_* by 44.9%, 75.7%, and 250.0%, and that of *T_r_* by 39.4%, 53.4%, and 138.0% under well-watered and water-stressed (70% and 30% FC) conditions, respectively.

### 2.5. Foliar-Applied MT Meliorates Chlorophyll Fluorescence Attributes under Water Deficit Stress

Compared to the unstressed plants, water-stressed plants had significantly lower values for the maximum quantum efficiency of PSII photochemistry (*F_v_*/*F_m_*), effective quantum yield of PSII photochemistry (*F_v′_*/*F_m’_*), actual photochemical efficiency of PSII (Φ_PSII_), electron transport rate (ETR), and photochemical quenching coefficient (*qP*). MT treatment significantly reduced water stress-induced declines in these attributes (Figure 5A–E). Water deficiency circumstances considerably induced the non-photochemical quenching coefficients (*qN*) value, which was mitigated via MT treatment (Figure 5F). It significantly (*p* < 0.05) provoked positive effects and improved the *F_v_*/*F_m_* by 43.0%, 69.9%, and 179.5%, *F_v′_*/*F_m’_* by 35.1%, 54.2%, and 146.7%, Φ_PSII_ by 44.8%, 82.9%, and 215.0%, ETR by 25.3%, 57.6%, and 172.5%, and that of *qP* by 22.1%, 47.3%, and 126.5% compared to values of the untreated plants under well-watered and water-stressed (70% and 30% FC) conditions, respectively.

### 2.6. Foliar Application of MT Ameliorates the Activity of Enzymes Involved in Photosynthetic Process under Water Scarcity

Barley plants grown under water stress had significantly lower activity of ribulose diphosphate carboxylase/oxygenase (Rubisco), fructose 1,6-bisphosphatase (FBPase), glyceraldehyde 3-phosphate dehydrogenase (GAPDH), and fructose 1,6-bisphosphate aldolase (FBA) than the unstressed ones. In contrast, MT foliar application significantly (*p* < 0.05) reduced water stress damage and boosted the activity of these enzymes (Figure 6A–D). Compared to the untreated plants at a severe (30% FC) water stress condition, MT treatment reduced the negative effects of water scarcity and significantly enhanced the activity of Rubisco by 161.4%, FBPase by 118.0%, GAPDH by 96.8%, and FBA by 72.8%.

### 2.7. Exogenously Applied MT Relives Water Deficiency-Induced Stress by Reducing Malondialdehyde (MDA) and Hydrogen Peroxide (H_2_O_2_) Production

The content of H_2_O_2_ and MDA was significantly elevated by water deficiency, while MT application significantly decreased their values under stressed conditions (Figure 7A,B). It significantly (*p* < 0.05) reduced the H_2_O_2_ content by 21.2%, 66.7%, and 112.0% and that of MDA content by 26.5%, 60.7%, and 100.0% compared to the untreated plants under well-watered and water-stressed (70% and 30% FC) circumstances, respectively.

### 2.8. The MT Foliage Application Modifies Cell Membrane Stability Index (MSI) and Lessen Electrolyte Leakage (EL) under Water Scarcity

Barley leaves’ MSI decreased under increasing water stress compared to the unstressed ones, while EL increased. In contrast, MT application increased barley’s ability to withstand water stress by raising MSI and lowering EL (Figure 8A,B). In comparison to the untreated plants, MT treatment significantly (*p* < 0.05) increased the MSI by 110.7%, while decreasing the EL by 33.3% in barley leaves under severe (30% FC) water stress conditions.

### 2.9. Exogenously Applied MT Improves Antioxidant Enzyme Activity under Water-Stressed Environments

The activities of superoxide dismutase (SOD), catalase (CAT), ascorbate peroxidase (APX), and glutathione reductase (GR) were all shown to be raised under water deficiency conditions, while monodehydroascorbate reductase (MDHAR) and dehydroascorbate reductase (DHAR) activity were found to be decreased. Curiously, all of their activities were significantly (*p* < 0.05) improved via MT treatment when water was scarce (Figure 9A–F). MT application under water-stressed (70% and 30% FC) conditions significantly increased the activity of SOD (36.2% and 74.4%), CAT (33.4% and 52.6%), APX (36.7% and 61.3%), MDHAR (66.7% and 164.5%), DHAR (51.9% and 129.2%), and GR (51.2% and 89.9%), respectively, relative to the control plants.

### 2.10. Foliar Application of MT Stimulates Endogenous Phytohormone Production under Water-Stressed Environments

In view of the effect of different irrigation treatments on endogenous plant hormone production, water scarcity resulted in a significant increase in salicylic acid (SA), melatonin (MT), and abscisic acid (ABA) concentration (Figure 10D–F) along with a significant decrease in indole-3-acetic acid (IAA), gibberellins (GAs), and cytokinins (CKs, trans-zeatin and trans-zeatin riboside) concentration (Figure 10A–C). However, in water-stressed MT-treated plants significant increases in IAA, GAs, CKs, SA, and MT acquisition, along with a significant decrease in ABA concentration, were detected (Figure 10A–F). MT application significantly (*p* < 0.05) increased the concentration of IAA by 20.3%, 58.2%, and 159.3%, GAs by 10.4%, 74.0%, and 181.9%, CKs by 12.3%, 61.7%, and 128.5%, SA by 12.3%, 61.7%, and 128.5%, and that of MT by 12.3%, 61.7%, and 128.5% compared to the untreated plants under well-watered and water-stressed (70% and 30% FC) conditions, respectively. Conversely, it significantly decreased ABA concentration by 12.3%, 61.7%, and 128.5% compared to the untreated plants under well-watered and water-stressed (70% and 30% FC) conditions, respectively.

## 3. Discussion

One of the main risks to the safety of the world’s food supply is water scarcity. Actually, plant morphological, physiological, and metabolic processes incur significant damage [1,2,5]. It has a negative impact on plant stomatal opening, transpiration, water content, cellular membrane structure, and endogenous phytohormone profile [10,47]. The improvement of drought-tolerant plant species is a global concern since the world population continues to rise and water resources for food production are depleting [12]. Indeed, a recent development utilized to increase water stress tolerance is exogenous treatment with natural substances like MT. It has been suggested that MT is a great candidate for crop enhancement since it can act as a stress-relieving agent [23,24,33,48,49]. A number of plant species, including Malus [42], *Medicago sativa* [15], wheat [50], creeping bentgrass [51], maize [25,34,52,53], rapeseed [28], *Davidia involucrata* [32], loquat [3], soybean [5,26], tobacco [46], tomato [30], and cotton [11,54], have shown that exogenous application of MT improves water stress tolerance. Nevertheless, our understanding of the mechanisms underlying MT-mediated barley tolerance to water constraint is currently limited. The current study, as a first investigation, reveals that MT foliar spray can increase barley’s ability to withstand water stress by enhancing endogenous hormonal state, photosynthetic efficiency, antioxidant activity, and cellular membrane integrity. This work provides further insight into the processes of water stress relief by barely using MT exogenous treatment.

The greatest danger that considerably lowers plant development and production is water deficiency [1,50,53,55]. The barley plants in the current study were cultivated in water-scarce conditions, which showed a considerable decline in their growth and productivity. This reduction could be explained by the consequences of ROS accumulation, which can harm significant cellular compartments and interfere with critical metabolic activities, including photosynthesis and antioxidant capacity [5,15,25,28]. Additionally, this investigation found that under water stress, photosynthetic pigment content, photochemical processes, Calvin cycle enzyme activity, photosynthetic rate, acquisition of essential phytohormones, and cell membrane structure all significantly decreased, which in turn lessens barley productivity. Contrarily, this approach displays that when water was scarce, MT application dramatically increased barley growth and yield. According to the obtained results, MT-mediated improved barley productivity in water-scarce conditions can be ascribed to reduced photosynthetic pigment degradation, increased photosynthetic rate, overcome stomatal limitations, activated photosynthetic enzymes, improved endogenous hormonal acquisition, altered biological membrane structure, and reduced oxidative damage by suppressing H_2_O_2_ production and activating antioxidant enzymes. This outcome is consistent with earlier research, which discovered that MT considerably enhanced plant growth and development in the presence of water scarcity [5,6,11,15,30,42,47,52]. These restorations in MT-treated plants are most likely the result of improved cell elongation and water content, reduced osmotic stress, as well as enhanced plant antioxidant activity [24,25,32,48]. In addition, it has been proposed that one of MT’s most particular auxin-like activities is its ability to promote growth [9]. Furthermore, increasing CKs content with MT treatment could greatly improve water-stressed plant development by delaying the induced senescence process, as CKs can act as a senescence retardant agent and plays an important role in plant photosynthesis [9,56]. In conclusion, the above findings showed that plant physiological features and reduction of ROS overproduction aided MT’s positive contributions to drought tolerance.

A damaging environmental stressor that impairs photosynthesis is considered to be a lack of water [10,53]. In low-water soils, plants block their stomata to decrease water loss, which reduces the amount of CO_2_ available for photosynthesis [25,28]. The results of the current investigation showed that a lack of water had an adverse effect on the concentration of photosynthetic pigments, the activity of photochemical reactions, the capacity of gas exchange, the chlorophyll fluorescence system, and the Calvin cycle enzyme activity, which in turn reduced the efficiency of photosynthetic activity. Most studies have shown that photosynthetic pigment content can decrease as a result of water stress-induced oxidative damage caused by increased ROS formation [6,10,25,26]. Additionally, dryness caused a considerable decline in gas exchange and chlorophyll fluorescence properties [10,11] as well as a reduction in PSII activity [34,57,58]. Another study proposed that a poor photosynthetic rate could be caused by a considerable suppression of Calvin cycle enzyme activity in stressed plants’ leaves [59]. Rubisco, GAPDH, FBPase, and FBA are the main enzymes involved in the Calvin cycle, which is crucial for controlling plant growth and development, as well as abiotic stress response. Increased FBA activity can assist in the Calvin cycle’s assimilation of CO_2_ [60], whereas Rubisco, GAPDH, and FBPase can stimulate carbon fixation, reduction, and RuBP regeneration, respectively [61]. In contrast, under water-stressed circumstances, the results of this study showed a considerable improvement in photosynthetic parameters in plants that had been foliar sprayed with MT. Improved water and nutrient absorption [11,28,33], alleviation of photoinhibition of PSII [62], and preservation of cell turgor, osmolytes level, membrane integrity, and antioxidant capacity are all attributed to MT-mediated improved photosynthetic attributes under water scarcity, suggesting that it has the potential to be a stress-relieving agent [17,18,24,37,42,52]. Additionally, under water stress conditions, exogenous MT treatment increased MT content, which can prevent chlorophyll degradation and downregulate the expression of chlorophyll catabolic enzymes and senescence-associated genes [11,51]. Collectively, MT treatment may improve leaf photosynthetic machinery and increase water stress tolerance in barley.

This study’s findings also showed that barley plants under water stress had oxidative damage, as seen by greater levels of H_2_O_2_, MDA, and EL as well as reduced MSI value. Similar outcomes are reported by [2,6,10,49,63]. Intriguingly, this experiment displayed a significant decline in the buildup of H_2_O_2_ and MDA in stressed plants treated with MT. The observed greater MSI and lower EL values that were detected in stressed treated plants could be explained by decreased MDA and H_2_O_2_ generation. Numerous studies have discovered that MT-sprayed plants exposed to stressful situations have decreased MDA, H_2_O_2_, and EL values and increased MSI levels, which is consistent with my findings [5,6,35,42,52]. Exogenously applied MT appeared to mitigate the effects of water stress-induced cellular damage and the ensuing loss of membrane integrity by reducing both H_2_O_2_ production and lipid peroxidation. Moreover, it was proposed that MT aids in the stabilization of biological membranes under challenging circumstances by preserving optimal fluidity [48] and controlling the expression of the lipid peroxidase genes [64]. Additional proof indicates MT regulates the antioxidant system to maintain H_2_O_2_ homeostasis [3,19,22,26,65]. According to credible evidence, MT is a natural free radical scavenger and a potent antioxidant that can counteract the effects of free radicals, assisting plants in maintaining redox homeostasis and reducing the effects of environmental stresses [23,34]. Similarly, MT has been demonstrated to be a powerful antioxidant at capturing and scavenging ROS [19,26,66]. Furthermore, increased carotenoid concentration in the leaves of stressed plants is credited with MT-mediated higher antioxidant capability during drought. Carotenoid, a substantial non-enzymatic compound, can lessen the accumulation of ROS and increase the drought tolerance of plants [4,10]. Additionally, this approach displays how MT dramatically increased the activity of key antioxidant enzymes, including SOD, CAT, APX, MDHAR, DHAR, and GR. This suggests that MT could enhance the ROS scavenging system and lessen the harm that water stress causes to cell membranes. These results correspond to those found by [11,24,32,37,67]. With regard to this issue, MT has been demonstrated to operate as a free radical scavenger, eliminating ROS and enhancing the activity of antioxidant enzymes via the control of gene transcription levels [65]. Based on the findings, the important roles of MT in scavenging free radicals and upregulating the activity of antioxidant enzymes appear useful in lowering oxidative damage induced by ROS. Furthermore, it is worth noting that the exogenous MT application can prevent EL, stabilize membranes, activate antioxidant enzymes, and maintain ROS levels within normal ranges, all of which help to preserve normal membrane function and lessen the deleterious consequences of water-scarce environments.

In order to withstand abiotic stress and control crop development processes, plant hormones are crucial [8,55,68]. The results of the current experiment showed that whereas SA, MT, and ABA content increased, IAA, GAs, and CKs levels were dramatically decreased as a result of water scarcity. This outcome is consistent with earlier research that revealed that in plants grown under stressful environments, IAA, GAs, and CKs content decreased, while SA, MT, and ABA content increased [6,10,14,41]. According to previous studies, the upregulation of ABA synthesis genes and signaling transcription factors under abiotic challenges could be the cause of the increased amount of endogenous ABA in plants [41,69]. On the contrary, this study’s findings showed that plants treated with MT under water stress acquired much more endogenous IAA, GAs, CKs, SA, and MT than the untreated ones. These results were consistent with earlier research that indicated MT treatment changes the profile of phytohormones [6,26,41,42,43,44]. This happens primarily as a result of the role that MT plays in relieving the effects of drought injury due to an intriguing interplay between MT and other plant hormones [9,11,16]. Furthermore, as shown by [65], the lowered H_2_O_2_ and MDA levels caused by MT treatment in water-stressed plants can control the homeostasis of endogenous plant hormones. Additionally, some studies believe that exogenous applied MT to stressed plants can upregulate the expression of biosynthesis enzymes, receptors, and elements of hormone signaling pathways [6,70], enhance the endogenous IAA content by prompting IAA biosynthesis [26,46], increase the endogenous GAs amount by upregulating GA biosynthesis genes [3], improve the endogenous CKs content by upregulating CKs biosynthesis genes (*LpIPT2* and *LpOG1*) and CKs signaling transcription factors (A-ARRs and B-ARRs) [41], and promote the endogenous SA level by activating glucose-6-phosphate dehydrogenase, shikimate dehydrogenase, and phenylalanine ammonia-lyase [71], which cause significant increases in internal phytohormonal acquisition. In a similar vein, it has been proposed that exogenous MT treatment can enhance endogenous hormone content in water-stressed plants by increasing the expression of a number of key genes involved in phytohormone signaling, particularly the IAA and GA signals [36]. In contrast, the present investigation shows that MT foliar spray dramatically reduced ABA buildup in plants under water stress. This may be caused by the action of MT on the regulation of ABA metabolism, which upregulates ABA catabolic genes and downregulates ABA biosynthesis genes [26,41,42,44,46]. It’s intriguing to note that using MT may have a considerable impact on endogenous phytohormone levels by controlling their biosynthesis and/or catabolism by altering the expression of their metabolism genes [3,26,41,46,70]. Additionally, it appears that modifying the phytohormonal profile in plants that have received MT treatment and are under water scarcity helps to improve the growth and development of barley and increase its tolerance to water stress.

Plant hormones modulate plant photosynthetic activity in response to abiotic stress [10,11]. In fact, ABA is the key hormone that controls plant water content by altering stomatal conductance in response to challenging conditions [10,14,55]. Similarly, it has been claimed that ABA serves as the core regulatory route in plants’ responses to drought, influencing leaf stomatal closure and activating the antioxidant system to reduce drought stress [72]. The current study found that when water was scarce, both net photosynthetic rate and stomatal conductance were decreased, implying that the drop-in photosynthetic rate was caused by stomatal closure. On the contrary, the findings of this study revealed that foliar application of MT mitigated the detrimental impacts of water shortage and dramatically enhanced photosynthetic rate and stomatal conductance values. This corresponds to the findings of [11,17,33,42,51]. This improvement could be attributed to the impact of MT, which lowers the levels of ABA and H_2_O_2_ in stressed plants treated with MT, enhancing stomatal function and mitigating stressor conditions, as was also discovered by [47]. Furthermore, it has been demonstrated that MT enhances stomata function by allowing stressed plants to reopen their stomata [52]; this reopening of stomata may be caused by MT treatment’s modulation of ABA content. It is worth noting that MT treatment reduces ABA levels and ROS burst, which mitigates water stress-induced senescence. Additionally, this study hypothesizes that MT-induced modification of endogenous hormone production and stimulation of the photosynthetic process may be the reason for the amelioration of the water stress in barley plants.

The current study’s findings also demonstrate that water scarcity dramatically boosted the endogenous level of MT in barley leaves compared to the non-stressed controls, suggesting that it may have served as a stress-relieving agent in these challenging circumstances. Actually, the abiotic stress-induced ROS burst may be responsible for the increase in endogenous MT levels by upregulating the MT biosynthesis genes [6,65]. Additionally, earlier studies hypothesized that environmental factors cause high levels of endogenous MT by altering the expression of stress-responsive genes and regulation factors that mitigate the negative effects of abiotic stressors on physiological processes [9,19]. Under stressful circumstances, there was an upregulation of the expression of the MT biosynthesis genes (T5H, tryptophan 5-hydroxylase; TDC, tryptophan decarboxylase; SNAT, serotonin *N*-acetyltransferase; and HIOMT, hydroxyindole-*O*-methyltransferase) [42]. Hence, as MT may interact with other hormones or signaling molecules to exert its beneficial effects on stress tolerance, stress also triggers endogenous MT production [26,41]. Interestingly, the present study manifested a significant increment in the endogenous MT content via MT foliar application under both non-stressed and stressed circumstances. This improvement could be explained by the effect of exogenous MT on triggering endogenous MT biosynthetic activity and thus plant water stress tolerance, which could be accomplished by increasing the expression of MT biosynthesis genes [11,73] and the level of *TaSNAT* transcript, which encodes a key regulatory enzyme in the MT biosynthetic pathway [74].

Based on the findings of this study, exogenously applied MT increased endogenous MT content that supports barley’s ability to withstand water stress by improving plant growth and production via enhancing photosynthetic efficiency, reprogramming antioxidant capacity, and triggering endogenous phytohormone production (Figure 11).

## 4. Materials and Methods

### 4.1. Plant Material and Growing Condition

Seeds for barley (*Hordeum vulgare* L. cv. Giza 123) were kindly provided by the Agriculture Research Center, Egyptian Ministry of Agriculture. Due to the Giza 123 cultivar’s high yield production, it was chosen. Melatonin foliar spray was used to boost barley water stress tolerance. Plastic pots with a 30 cm diameter and a 35 cm depth were used to plant the seeds. Fifteen kg of clay loamy soil (sand 37%, silt 28%, clay 35%) was used to fill the pots. Table 1 shows the results of a soil chemical analysis that was performed in accordance with the instructions of [75]. The pots were set up in a greenhouse at the Department of Plant Physiology, Faculty of Agriculture, Cairo University, Egypt, where they were exposed to natural light and temperature conditions, with an average day/night temperature of 22/16 ± 2 °C and an average humidity of 65%. On 10 September 2021 and 2022, the experiment was conducted twice with consistent results.

Pots were separated into three groups before sowing. Well-watered (WW; 100% of field capacity) served as the first group’s control, while the other two groups represented two levels of irrigation treatments (WD1; 70% of field capacity) and (WD2; 30% of field capacity). According to the formula, soil water content (SWC) % = [(fresh weight (FW) − dry weight (DW)/DW] × 100 [76], the SWC was 15.5%, 10.8%, and 4.6%, respectively. All irrigation levels were maintained from grain sowing to grain filling.

At each irrigation level, barley plants at 45 (stem elongation stage), 60 (booting stage), and 90 (heading stage) days from sowing were subjected to two MT treatments: treated with MT and untreated plants. Untreated plants were sprayed with distilled water. According to preliminary investigation, 70 μM MT was the most effective concentration among a range of MT concentrations ranging from 0 to 100 μM. Melatonin was acquired from Sigma (USA) and dissolved in an appropriate amount of ethanol. Tween-20 (0.05%) was used as a surfactant during treatment. The recommended fertilization and maintenance techniques were followed.

The experiment was set up in a completely randomized design, with two factors (three levels of irrigation and two foliar spraying treatments) and four replicates.

### 4.2. Growth Evaluation

Plant height, leaf number, total leaf area, and shoot dry weight were measured on 75-day-old plants. A portable leaf area meter (LI-COR 3000, Lambda Instruments Corporation, Lincoln, NE, USA) was used to estimate total leaf area. After 48 h of oven drying at 70 °C, the dry weight of the shoot was measured. Each treatment has four replicates, and each replicate has six plants taken from the same pot.

### 4.3. Plant Productivity Analysis

At maturity, number of grains and grain yield plant^−1^ were recorded.

The following physiological and biochemical traits were determined in 75-day-old barley leaves. Four replicates were used to obtain the data, and each one had six plants that were taken from the same pot.

### 4.4. Photosynthetic Pigments Measurement

According to the procedure of [77], photosynthetic pigments from leaves were extracted in 80% (*v*/*v*) acetone, and the quantities of chlorophyll *a*, chlorophyll *b*, and carotenoids were measured spectrophotometrically using a UV-1750 spectrophotometer (Shimadzu, Kyoto, Japan).

### 4.5. Chloroplast Isolation and Measurement of Photosynthetic Photochemical Reactions Activity

Isolating the chloroplast was accomplished using the method described by [78]. PSII-mediated electron transfer from H_2_O to *p*-benzoquinone was discovered, according to [79]. PSI-mediated electron transport was evaluated in terms of oxygen consumption using 2, 6-dichlorophenol indophenols as the electron donor and methyl viologen as the final acceptor [80].

### 4.6. Gas Exchange Measurement

The gas exchange of attached leaves was measured at 8:30–11:30 am using an infrared gas analyzer, Li-Cor-6400 (Li-Cor Inc., Lincoln, NE, USA) equipped with an LED red/blue light source (6400-02B). The photosynthetic photon flux density (PPFD) was set at 1000 µmol m^−2^ s^−1^, and the cuvette air flow rate was 500 mL min^−1^. Net photosynthetic rate (*P_n_*, μmol CO_2_ m^−2^ s^−1^), stomatal conductance (*G_s_*, mol H_2_O m^−2^ s^−1^), and transpiration rate (*T_r_*, mmol H_2_O m^−2^ s^−1^) were recorded simultaneously.

### 4.7. Chlorophyll Fluorescence Measurements

Chlorophyll fluorescence parameters were measured in barley leaves with a Portable Chlorophyll Fluorometer (PAM2500; Heinz Walz, Effeltrich, Germany) after a 30 min dark adaptation. The maximum quantum efficiency of PSII photochemistry (*F_v_*/*F_m_*), electron transport rate (ETR), actual photochemical efficiency of PSII (Φ_PSII_), photochemical quenching coefficient (*qP*), effective quantum yield of PSII photochemistry (*F_v′_*/*F_m’_*), and non-photochemical quenching coefficients (*qN*) were calculated according to [81].

### 4.8. Assay of Calvin Cycle Enzymes

Calvin cycle enzymes [ribulose diphosphate carboxylase/oxygenase (Rubisco), fructose 1,6-bisphosphatase (FBPase), glyceraldehyde 3-phosphate dehydrogenase (GAPDH), and fructose 1,6-bisphosphate aldolase (FBA)] activity was assessed using ELISA kits (Yaji Biotech, Shanghai, China). In an extraction buffer containing 0.05 mM Tris-HCl and 0.1 M phosphate buffer, pH 7.4, barley leaf samples were ground. They were then under centrifugation procedure (3000× *g* for 15 min at 4 °C). The enzyme activity assay was conducted using the supernatant. The test sample (the standard and the horseradish peroxidase-conjugate reagent) was placed in the microplate wells. The Rubisco antibody was already present in the microplate wells for determining Rubisco activity. After incubating for 60 min at 37 °C, the samples were washed. Peroxidase transformed the substrate 3,3′,5,5′- tetramethylbenzidine to blue, and acid action converted it to yellow. At 450 nm, the color’s intensity was determined. The optical density of the samples was then compared to the standard curve to assess Rubisco activity. Other enzymes’ activities were also measured in the same way. Enzyme activity (U) is defined as the quantity of enzyme required to convert 1 µmol of substrate in 1 min under optimal conditions.

### 4.9. Determination of Hydrogen Peroxide (H_2_O_2_) and Malondialdehyde (MDA)

According to the procedure given by [82], 0.1 g of fresh barley leaves were ground in a mortar with 900 µL of buffer for the determination of H_2_O_2_ and MDA. The instructions provided in the H_2_O_2_ and MDA kits were followed. At 405 nm and 532 nm, respectively, the concentrations of H_2_O_2_ and MDA were measured.

### 4.10. Extraction and Assay of Antioxidant Enzymes

Fresh leaves from barley (0.5 g) were homogenized in 5 mL of ice-cold 100 mM phosphate buffer (pH 7.4) with 1% polyvinyl pyrrolidine and 1 mM EDTA before being centrifuged at 15,000× *g* for 10 min at 25 °C. In order to conduct the enzymatic analyses, the supernatant was gathered. The SOD activity was assessed using the approach of [83] by monitoring its inhibition of the photochemical reduction of nitroblue tetrazolium. One unit of enzyme activity was defined as the amount of enzyme that inhibited the reduction rate of nitroblue tetrazolium by 50% when measured at 560 nm. The CAT activity was measured by measuring the drop in absorbance at 240 nm caused by H_2_O_2_ breakdown [84]. According to [85], APX activity was determined by measuring the decrease in absorbance at 290 nm caused by ascorbate oxidation. The MDHAR activity was measured using NADH oxidation at 340 nm [86]. The DHAR activity was determined by observing ascorbate production at 265 nm [87]. The GR activity was measured by detecting NADPH oxidation at 340 nm [88].

### 4.11. Estimation of Electrolyte Leakage (EL) and Membrane Stability Index (MSI)

To investigate ion leakage from membranes, barley leaves were collected and rinsed in distilled water. They were placed in test tubes containing 10 mL of distilled water and placed in a water bath at 40 °C for 30 min, during which time electrical conductivity (C1) was measured. The identical samples were then immersed in a 100 °C water bath for 10 min to measure electrical conductivity (C2). Using the formula of [89], the electrolyte leakage was estimated. EL = [C1/C2] × 100

The MSI was calculated using the approach described in [90]. Barley leaf samples (0.2 g) were incubated in 10 mL of deionized water and gently shaken at room temperature for 24 h. The conductance was measured after incubation and recorded as C1. Following that, leaf samples were incubated at 120 °C for 20 min before being shaken for 24 h at room temperature. C2 is the final conductance that was measured. MSI for leaves was determined using the following formula: MSI = [1 − C1/C2]× 100

### 4.12. Extraction and GC–MS Analysis of Endogenous Phytohormones

With some modifications, the endogenous hormones, including IAA, CKs, GAs, SA, and ABA, were measured in accordance with [91]. Briefly, 2 mL of ice-cold extraction solvent (methanol/water/HCl (6N); 80/19.9/0.1; *v*/*v*/*v*) was used to extract one hundred mg of dried ground leaves after they had been homogenized and ground in liquid nitrogen. The supernatant was collected and used to determine the presence of phytohormones after the extract had been centrifuged at 25,000× *g* for 5 min at 4 °C. For IAA, ABA, and SA, 50 μL of the supernatant was derivatized with 40 μL of methyl chloroformate (MCF) and dried by adding sodium sulfate. For CKs and GAs, 50 μL of the supernatant was derivatized and dried with 100 μL of *N*-Methyl-*N*-(trimethylsilyl) trifluoroacetamide (MSTFA) by heating at 85 °C for 45 min. For GC–MS analysis, 1 μL was injected into the GC–MS running in the selective ion mode (SIM-mode). A Clarus 680 GC with SQ8-T Mass Spectrometer system (Perkin Elmer, Waltham, MA, USA) fitted with an Elite-5MS capillary column (low bleed, 30 m × 0.25 mm × 0.025 μm film thickness; Perkin Elmer, Waltham, MA, USA) was used to analyze all samples and phytohormone standards. Helium was the carrier gas with flow rate of 1 mL min^−1^. For IAA, ABA, and SA, the temperature program was as follows: the column was maintained at 50 °C for 3 min and then was raised to 200 °C at a rate of 4 °C min^−1^, held for 5 min. The procedure for GAs and CKs was as follows: the column was kept at 60 °C for 2 min, increased to 160 °C at 20 °C min^−1^, and then increased to 290 °C at 5 °C min^−1^. Temperatures for the injector and detector were set at 250 °C and 260 °C, respectively. The TurboMass software version 6.1 (Perkin Elmer, Waltham, MA, USA) was used to analyze chromatograms. Identification of IAA, ABA, SA, GAs, and CKs was performed by comparing their retention time, linear retention indices (LRIs), and the selected ions with those of authentic standards.

### 4.13. Melatonin Extraction and Analysis

The extraction and measurement of endogenous MT in barley leaves were carried out using the method reported in [92]. Briefly, 0.5 g fresh leaf sections were crushed with liquid nitrogen before being extracted with 5 mL chloroform overnight at 4 °C. The chloroform phase was evaporated after centrifuging at 10,000× *g* for 15 min at 4 °C. One mL of 42% methanol was used to dissolve the dry residue. High-Performance Liquid Chromatography (HPLC) with fluorescence detection was used to evaluate 400 mL aliquots. The samples were separated on a Shim-pack VP-ODS column (3 mm, 4.6 × 150 mm, Shimadz × 2% methanol to 50% methanol in 0.1% formic acid for 27 min, then isocratic elution with 50% methanol in 0.1% formic acid for 18 min at a flow rate of 0.15 mL/min). The excitation wavelength was 280 nm, and the emission wavelength was 348 nm.

### 4.14. Statistical Analysis

The data was analyzed using two-way variance analysis (ANOVA). The data were evaluated using a completely randomized approach with four replications. Because the outcomes of the two seasons followed a similar pattern, a combined analysis was performed for them. The least significant difference (LSD) test was used to estimate the statistical significance of the means at *p* < 0.05. For data analysis, SAS software (release 9.1, SAS Inc., Cary NC, USA) was used. The data are presented as means ± standard error (SE).

## 5. Conclusions

Water stress tolerance is an important trait for sustainable crop production. The present study demonstrates that exogenously applied MT could enhance barley water stress tolerance by improving growth and productivity via reprogramming photosynthetic efficiency, antioxidant capacity, and endogenous phytohormone production. MT treatment alleviated the water stress-reduced photosynthesis, which would be particularly attributed to an increase in photosynthetic pigment content, photochemical reactions of photosynthesis, Calvin cycle enzyme activity, gas exchange capacity, and chlorophyll fluorescence system. Exogenous MT application also relived water stress-induced oxidative damage by increasing the plant capacity to eliminate H_2_O_2_, reduce membrane lipid peroxidation, lessen electrolyte leakage, and improve cell membrane stability via activating antioxidant enzymes (SOD, CAT, APX, MDHAR, DHAR, and GR). Furthermore, MT foliage application under water-scarce conditions enhanced the content of endogenous MT that can regulate the homeostasis of endogenous plant hormones by promoting the level of endogenous IAA, GAs, CKs, and SA, as well as inhibiting the internal ABA content. In conclusion, the results of this study revealed that MT can be regarded as a key protector from water stress and also provides further evidence for the physiological role of MT and a theoretical basis for its application in improving water stress tolerance in agricultural practice.

## Figures and Tables

**Figure 1 ijms-24-16228-f001:**
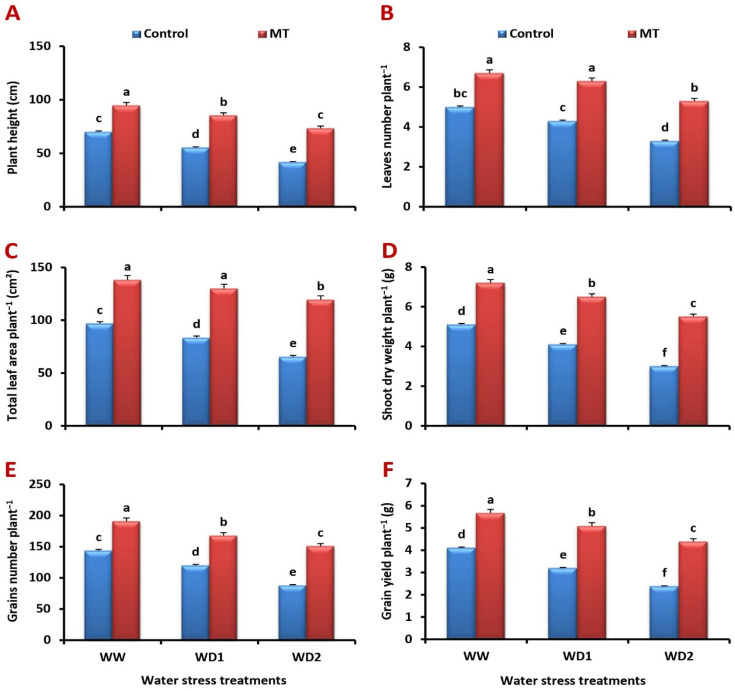
Influence of melatonin (MT) foliar application on the: (**A**) plant height (cm), (**B**) leaves number plant^−1^, (**C**) total leaf area plant^−1^ (cm^2^), (**D**) shoot dry weight plant^−1^ (g), (**E**) grains number plant^−1^, and (**F**) grain yield plant^−1^ (g) of barley plants grown under different water deficit conditions [well-watered (WW), water deficit stress (WD1; 70% of field capacity), and water deficit stress (WD2; 30% of field capacity)]. Vertical bars represent ±standard error of the mean (*n* = 4). Different letters indicate significant differences between treatments at *p* < 0.05 level according to LSD test.

**Figure 2 ijms-24-16228-f002:**
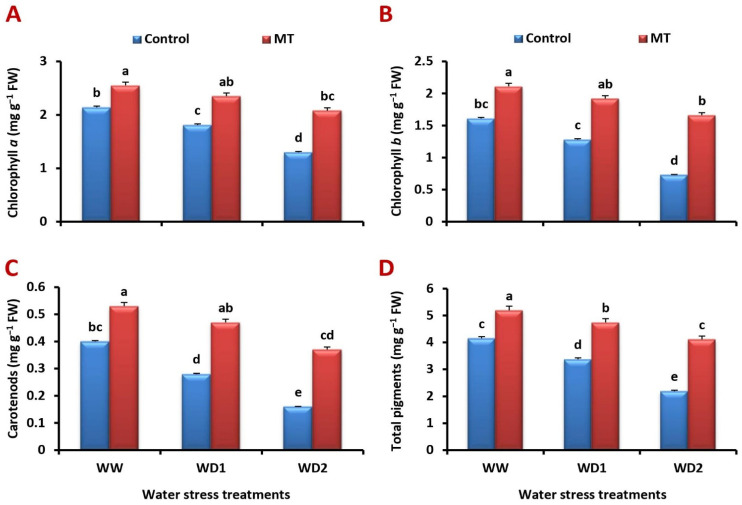
Influence of melatonin (MT) foliar application on the concentration of (**A**) chlorophyll *a*, (**B**) chlorophyll *b*, (**C**) carotenoids, and (**D**) total pigments (mg g^−1^ FW) in leaves of barley plants grown under different water deficit conditions [well-watered (WW), water deficit stress (WD1; 70% of field capacity), and water deficit stress (WD2; 30% of field capacity)]. Vertical bars represent ±standard error of the mean (*n* = 4). Different letters indicate significant differences between treatments at *p* < 0.05 level according to LSD test.

**Figure 3 ijms-24-16228-f003:**
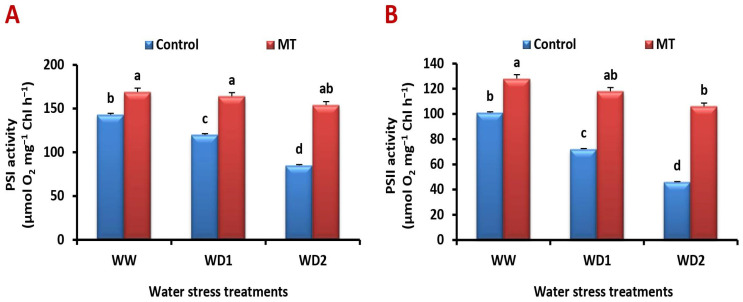
Influence of melatonin (MT) foliar application on the (**A**) PSI and (**B**) PSII electron transport activities (µmol O_2_ mg^−1^ Chl h^−1^) in leaves of barley plants grown under different water deficit conditions [well-watered (WW), water deficit stress (WD1; 70% of field capacity), and water deficit stress (WD2; 30% of field capacity)]. Vertical bars represent ±standard error of the mean (*n* = 4). Different letters indicate significant differences between treatments at *p* < 0.05 level according to LSD test.

**Figure 4 ijms-24-16228-f004:**
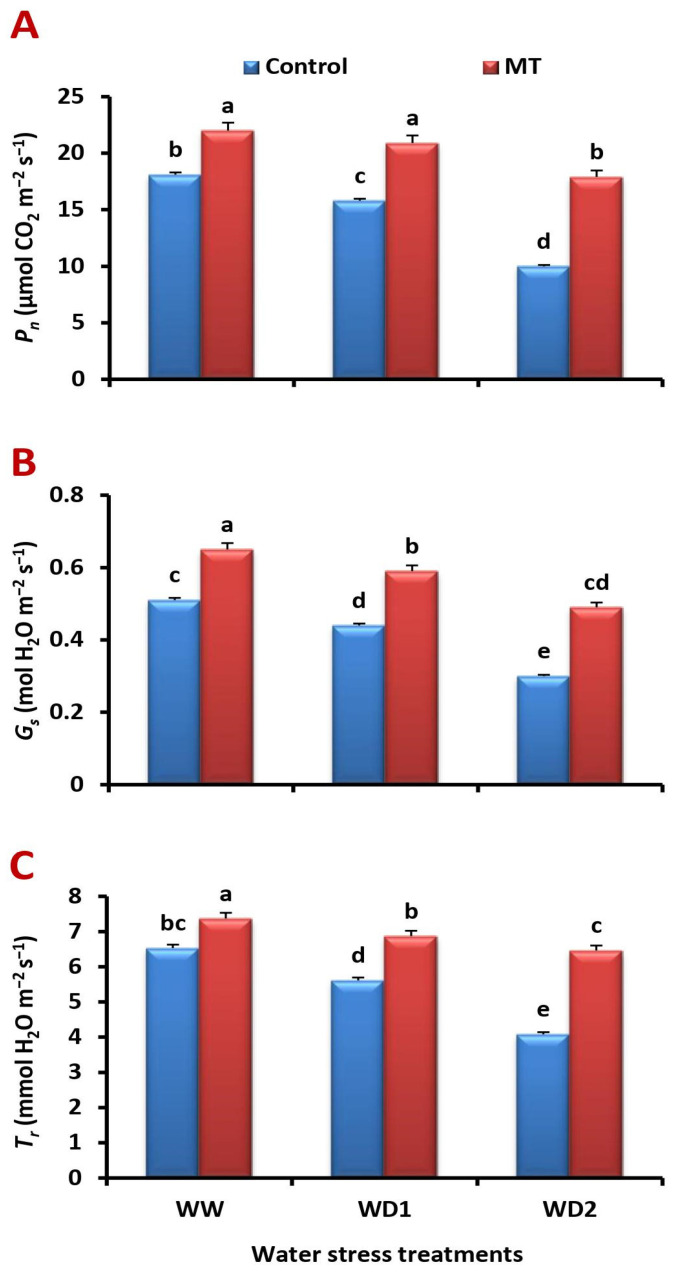
Influence of melatonin (MT) foliar application on the (**A**) net photosynthetic rate (*P_n_*, μmol CO_2_ m^−2^ s^−1^), (**B**) stomatal conductance (*G_s_*, mol H_2_O m^−2^ s^−1^), and (**C**) transpiration rate (*T_r_*, mmol H_2_O m^−2^ s^−1^) in leaves of barley plants grown under different water deficit conditions [well-watered (WW), water deficit stress (WD1; 70% of field capacity), and water deficit stress (WD2; 30% of field capacity)]. Vertical bars represent ±standard error of the mean (*n* = 4). Different letters indicate significant differences between treatments at *p* < 0.05 level according to LSD test.

**Figure 5 ijms-24-16228-f005:**
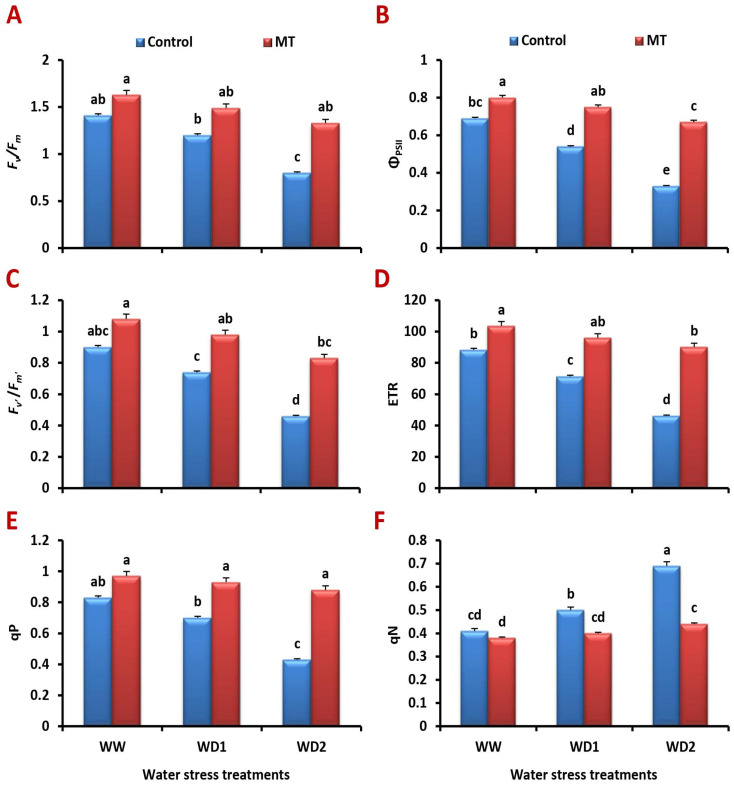
Influence of melatonin (MT) foliar application on the (**A**) maximum quantum efficiency of PSII photochemistry (*F_v_*/*F_m_*), (**B**) actual photochemical efficiency of PSII (Φ_PSII_), (**C**) effective quantum yield of PSII photochemistry (*F_v′_*/*F_m’_*), (**D**) electron transport rate (ETR), (**E**) photochemical quenching coefficient (qP), and (**F**) non-photochemical quenching coefficients (qN)] in leaves of barley plants grown under different water deficit conditions [well-watered (WW), water deficit stress (WD1; 70% of field capacity), and water deficit stress (WD2; 30% of field capacity)]. Vertical bars represent ±standard error of the mean (*n* = 4). Different letters indicate significant differences between treatments at *p* < 0.05 level according to LSD test.

**Figure 6 ijms-24-16228-f006:**
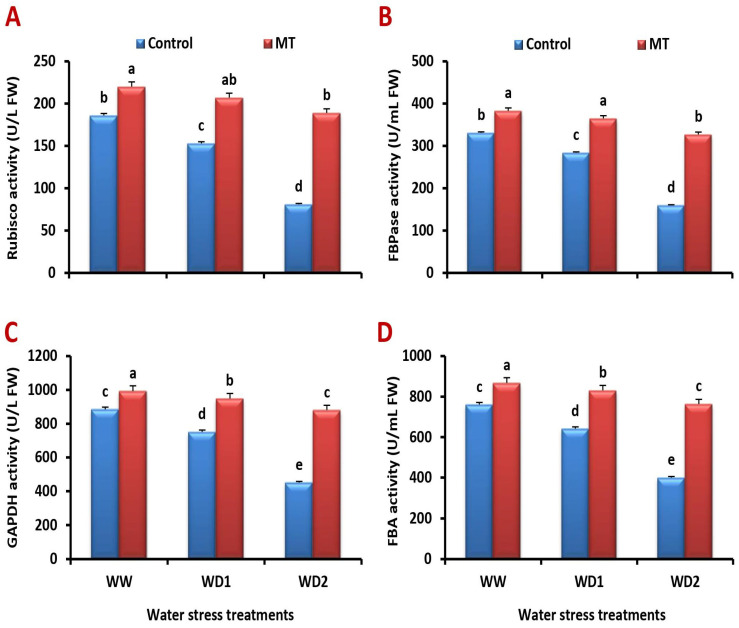
Influence of melatonin (MT) foliar application on the activity of (**A**) ribulose diphosphate carboxylase/oxygenase (Rubisco), (**B**) fructose 1,6-bisphosphatase (FBPase), (**C**) glyceraldehyde-3-phosphate dehydrogenase (GAPDH), and (**D**) fructose 1,6-bisphosphate aldolase (FBA) in leaves of barley plants grown under different water deficit conditions [well-watered (WW), water deficit stress (WD1; 70% of field capacity), and water deficit stress (WD2; 30% of field capacity)]. Vertical bars represent ±standard error of the mean (*n* = 4). Different letters indicate significant differences between treatments at *p* < 0.05 level according to LSD test.

**Figure 7 ijms-24-16228-f007:**
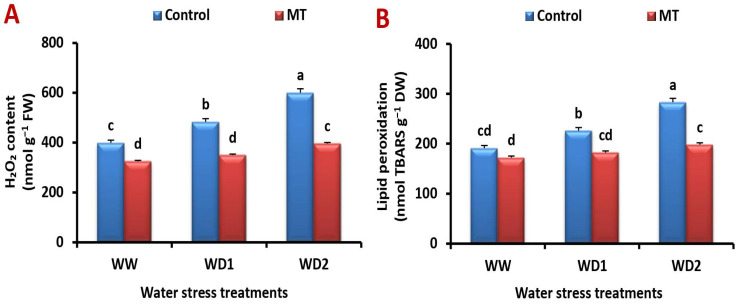
Influence of melatonin (MT) foliar application on the (**A**) hydrogen peroxide (H_2_O_2_) content and (**B**) lipid peroxidation in leaves of barley plants grown under different water deficit conditions [well-watered (WW), water deficit stress (WD1; 70% of field capacity), and water deficit stress (WD2; 30% of field capacity)]. Vertical bars represent ±standard error of the mean (*n* = 4). Different letters indicate significant differences between treatments at *p* < 0.05 level according to LSD test.

**Figure 8 ijms-24-16228-f008:**
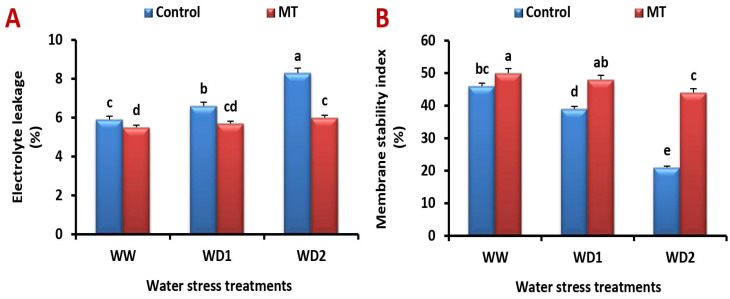
Influence of melatonin (MT) foliar application on the (**A**) electrolyte leakage (%) and (**B**) membrane stability index (%) in leaves of barley plants grown under different water deficit conditions [well-watered (WW), water deficit stress (WD1; 70% of field capacity), and water deficit stress (WD2; 30% of field capacity)]. Vertical bars represent ±standard error of the mean (*n* = 4). Different letters indicate significant differences between treatments at *p* < 0.05 level according to LSD test.

**Figure 9 ijms-24-16228-f009:**
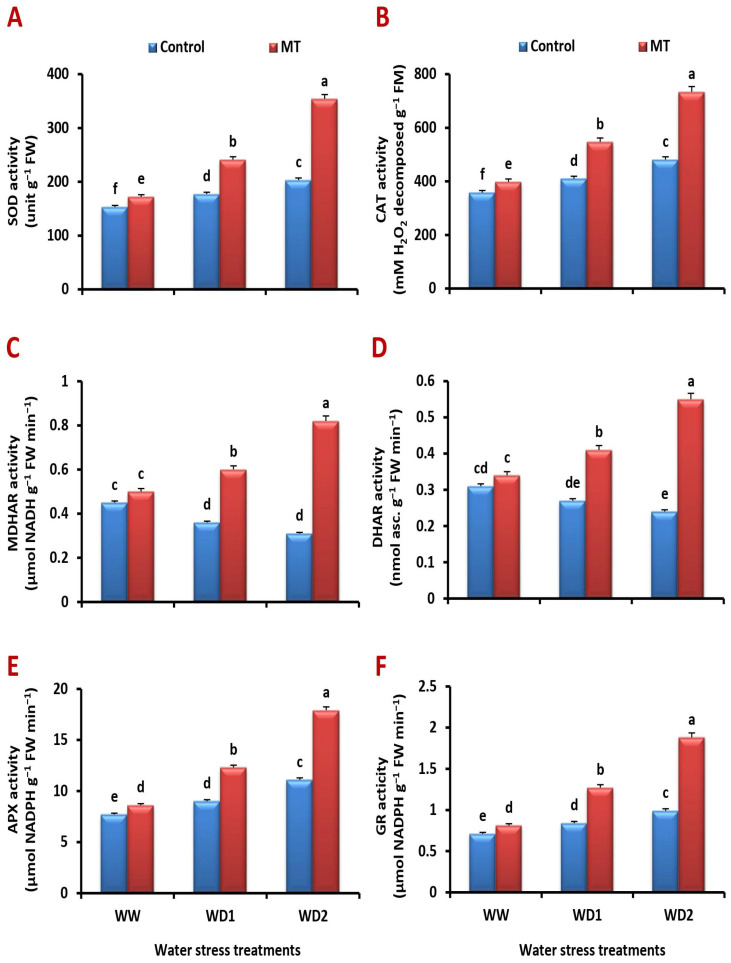
Influence of melatonin (MT) foliar application on the activity of (**A**) superoxide dismutase (SOD), (**B**) catalase (CAT), (**C**) ascorbate peroxidase (APX), (**D**) monodehydroascorbate reductase (MDHAR), (**E**) dehydroascorbate reductase (DHAR), and (**F**) glutathione reductase (GR) in leaves of barley plants grown under different water deficit conditions [well-watered (WW), water deficit stress (WD1; 70% of field capacity), and water deficit stress (WD2; 30% of field capacity)]. Vertical bars represent ±standard error of the mean (*n* = 4). Different letters indicate significant differences between treatments at *p* < 0.05 level according to LSD test.

**Figure 10 ijms-24-16228-f010:**
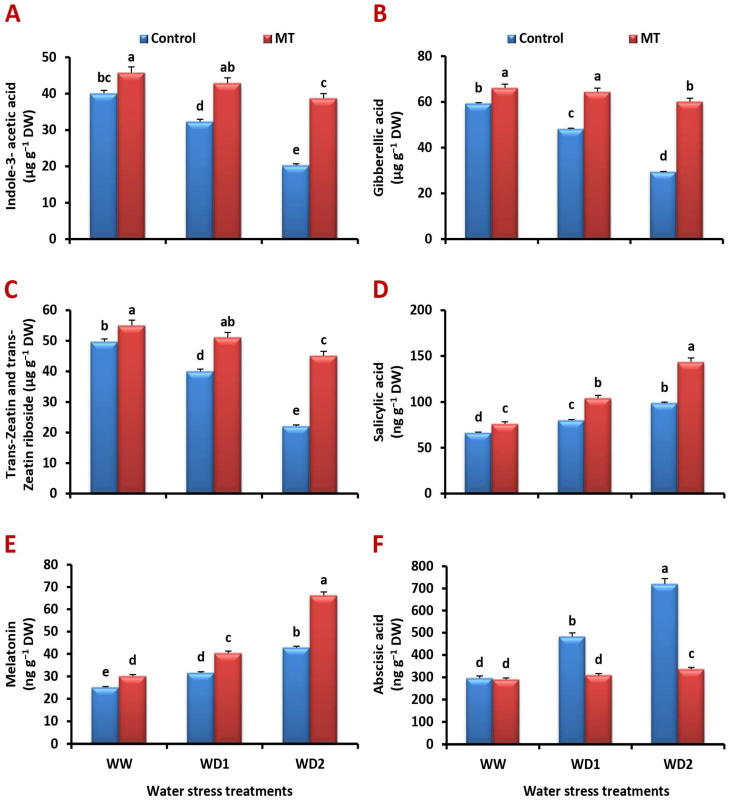
Influence of melatonin (MT) foliar application on the concentration of endogenous: (**A**) indole-3-acetic acid (µg g^−1^ DW), (**B**) gibberellins (µg g^−1^ DW), (**C**) cytokinins (µg g^−1^ DW), (**D**) salicylic acid (ng g^−1^ DW), (**E**) melatonin (ng g^−1^ DW), and (**F**) abscisic acid (ng g^−1^ DW) in leaves of barley plants grown under different water deficit conditions [well-watered (WW), water deficit stress (WD1; 70% of field capacity), and water deficit stress (WD2; 30% of field capacity)]. Vertical bars represent ±standard error of the mean (*n* = 4). Different letters indicate significant differences between treatments at *p* < 0.05 level according to LSD test.

**Figure 11 ijms-24-16228-f011:**
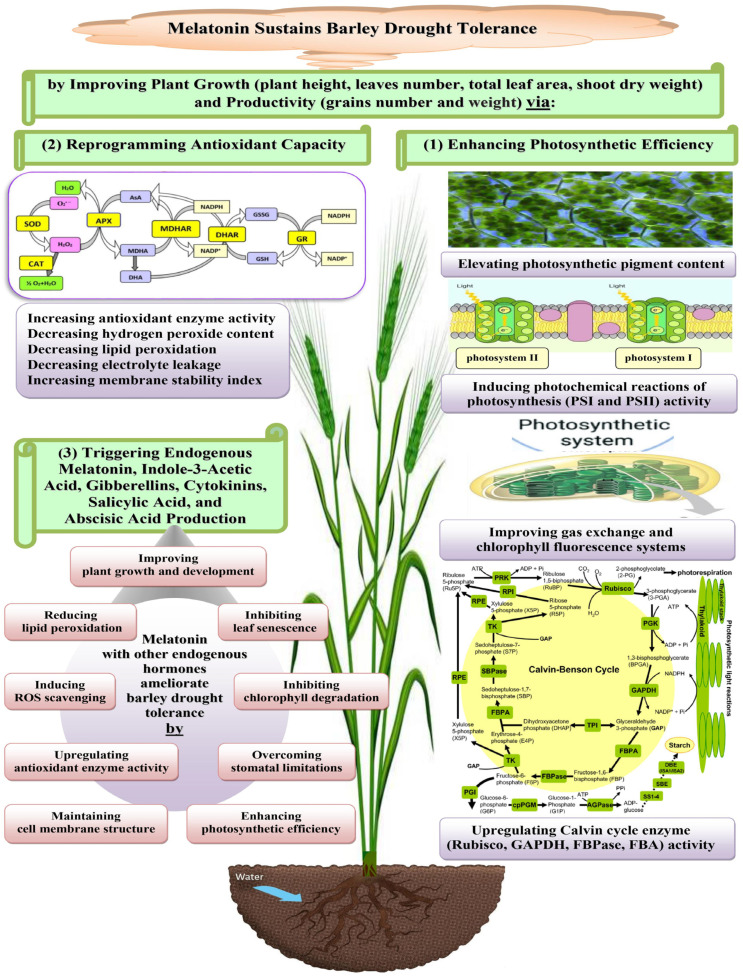
Melatonin (MT) foliar application sustains barley water stress tolerance by enhancing photosynthetic efficiency, reprogramming antioxidant capacity, and triggering endogenous phytohormone production.

**Table 1 ijms-24-16228-t001:** Basic characteristics of the soil used in this experiment.

pH	HCO_3_^−^ + CO_3_^2−^ (mg kg^−1^)	Cl^−^ (mg kg^−1^)	SO_4_^2−^ (mg kg^−1^)	Ca^2+^ (mg kg^−1^)	Mg^2+^ (mg kg^−1^)	Na^+^ (mg kg^−1^)	K^+^ (mg kg^−1^)	N (mg kg^−1^)	P (mg kg^−1^)
7.2	210.3	314.1	426.5	90.0	40.8	3.6	31.7	19.4	3.3

## Data Availability

Data are contained within the article.

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
