# Peer review of "Drought Stress Alleviator Melatonin Reconfigures Water-Stressed Barley (Hordeum vulgare L.) Plants’ Photosynthetic Efficiency, Antioxidant Capacity, and Endogenous Phytohormone Profile"

_ijms, 2023, doi:10.3390/ijms242216228_

Round 1
Reviewer 1 Report
Comments and Suggestions for Authors
Dear Author,
Author concentrated on exogenous melatonin (MT)-mediated drought tolerance in barley. Obtained results showed how MT can potentially mitigate changes in barley physio-biochemical parameters caused by drought.
Author presented interested new findings demonstrated that increased levels of SA, gibberellins, cytokinins, melatonin, and IAA, as well as a decrease in ABA, indicated that foliar-applied melatonin significantly improved barley drought tolerance. Additionally, by increasing the activity of antioxidant enzymes like: SOD, CAT, DHAR, MDHAR, APX and GR and simultaneously decreasing H202 content, lipid peroxidation, and electrolyte leakage, MT application decreased oxidative stress in drought condition.
Introduction gives the reader sufficient background also with working-hypothesis, but I suggest to add also precise aims of the study;
Results are well organized and are presented in a clear way with informative figures;
I did not find any supplementary data, Please think about adding plants “symptoms” after MT application, without in drought condition and control;
I suggest only introduce figure 3 to the manuscript properly;
In my opinion materials and methods section are properly constructed;
The discussion part is exhaustive and well guided, adding very nice scheme at the and was a very good choice;
Sincerely
Comments on the Quality of English LanguageMinor English editing correction is only needed;
Author Response
- Dear Reviewer #1: Thanks a lot for your valuable comments. As suggested, I have introduced figure 3 to the manuscript properly.
Reviewer 2 Report
Comments and Suggestions for Authors
The manuscript “Drought Stress Alleviator Melatonin Reconfigures Water-Stressed Barley (Hordeum vulgare L.) Plants' Photosynthetic Efficiency, Antioxidant Capacity, and Endogenous Phytohormone Profile” examines the effects of melatonin foliar spray application to barley cultivation, in terms of its growth response to water stress. The study evaluates several physiology and growth parameters of barley plants, under greenhouse experimental conditions in Egypt.
The paper is well-written and properly structured. The topic is timely and interesting, dealing with the sensitive matter of plant growth under water scarcity, which is crucial especially in regions with arid or semi-arid conditions. The introduction presents sufficiently the topic under study, while the methodological approach seems sound.
An issue that should be further clarified to avoid confusion or misinterpretations by the readers, is the use of ‘drought’, which is a main term used in the title and throughout the text. Drought is typically conceived as the natural (extreme) phenomenon related to the significant decrease of precipitation (thus available soil moisture), usually combined with the increase of temperature (thus increased potential evapotranspiration), compared to the normal conditions of a region. However, the present study actually deals with plant growth under water stress (deficit irrigation) under certain environmental conditions, which is related with water scarcity but not necessarily drought. Therefore, it is suggested to avoid using the term drought (throughout the text) and use consistently the terms water stress and/or water scarcity, instead. Otherwise, the term drought should be carefully and clearly defined in relation to the goals of the study, though this might still be confusing for the readers.
Overall, the paper has merit, presenting adequately the case under study.
Author Response
- Dear Reviewer #2: Thanks a lot for your valuable comments. As suggested, I omitted the term drought (throughout the text) and used consistently the terms water stress and/or water scarcity, instead.
Reviewer 3 Report
Comments and Suggestions for Authors
The manuscript "Drought Stress Alleviator Melatonin Reconfigures Water-Stressed Barley (Hordeum vulgare L.) Plants' Photosynthetic Efficiency, Antioxidant Capacity, and Endogenous Phytohormone Profile" delves into the beneficial aspects of using a melatonin spray for growing barley under conditions of more or less severe droughts.
The study is interesting and well conducted. The text of the manuscript is easy to read.
The following changes are suggested.
Line 23-24: Enter a percentage of the improvement
Figure 3: Figure 3B appears cropped. Please, reformat the figure
Line 544: Enter the type of melatonin foliar spray used
Line 559: Explicit FW and DW
Author Response
- Dear Reviewer #3: Thanks a lot for your valuable comments. As suggested, I have made the following changes.
- Line 23-24: Enter a percentage of the improvement.
Thank you for your comment. But it is very difficult because abstract should be not more 200 words.
- Figure 3: Thank you, I have reformatted the figure.
- Line 544: Enter the type of melatonin foliar spray used.
Thank you for your comment. It is already given in Lines 567-569.
- Line 559: Thank you, I have explicated FW and DW.